

# Effects of shokyo (*Zingiberis Rhizoma*) and kankyo (*Zingiberis Processum Rhizoma*) on prostaglandin E₂ production in lipopolysaccharide-treated mouse macrophage RAW264.7 cells

Toshiaki Ara[1], Masanori Koide[2], Hiroyuki Kitamura[3] and Norio Sogawa[1]

[1] Department of Pharmacology, Matsumoto Dental University, Shiojiri, Nagano, Japan
[2] Institute for Oral Science, Matsumoto Dental University, Shiojiri, Nagano, Japan
[3] Matsumoto Dental University Hospital, Shiojiri, Nagano, Japan

## ABSTRACT

We previously reported that shokyo and kankyo, which are water-extracted fractions of ginger, reduced LPS-induced $PGE_2$ production in human gingival fibroblasts. In this study, we examined the effects of these herbs on LPS-treated mouse macrophage RAW264.7 cells. Both shokyo and kankyo reduced LPS-induced $PGE_2$ production in a concentration-dependent manner. Shokyo and kankyo did not inhibit cyclooxygenase (COX) activity, nor did they alter the expression of molecules in the arachidonic acid cascade. In addition, these herbs did not alter NF-κB p65 translocation into nucleus, or phosphorylation of p65 or ERK. These results suggest that shokyo and kankyo inhibit $cPLA_2$ activity. Although 6-shogaol produced similar results to those of shokyo and kankyo, the concentration of 6-shogaol required for the reduction of $PGE_2$ production were higher than those of 6-shogaol in shokyo and kankyo. Therefore, several gingerols and shogaols other than 6-shogaol may play a role in the reduction of LPS-induced $PGE_2$ production. Thus, 6-shogaol, and other gingerols and shogaols inhibit $cPLA_2$ activity and reduce LPS-induced $PGE_2$ production via a different mechanism from traditional anti-inflammatory drugs. Moreover, kampo medicines that contain shokyo or kankyo are considered to be effective for inflammatory diseases.

Corresponding author
Toshiaki Ara, toshiaki.ara@mdu.ac.jp

## INTRODUCTION

Japanese traditional medicines (kampo medicines) are used for the treatment of several inflammatory diseases. We focused on the inflammatory effects of these kampo medicines, and found that lipopolysaccharide (LPS)-induced $PGE_2$ production by human gingival fibroblasts (HGFs) were reduced by several kampo medicines, including shosaikoto (TJ-9) (*Ara et al., 2008*), orento (TJ-120) (*Ara et al., 2010*), hangeshashinto (TJ-14) (*Nakazono et al., 2010*), kakkonto (TJ-1) (*Kitamura, Urano & Ara, 2014*), shinbuto (TJ-30), and ninjinto (TJ-32) (*Ara & Sogawa, 2017*). Moreover, among the herbs contained in

kakkonto, shokyo (*Zingiberis Rhizoma*), kanzo (*Glycyrrhizae Radix*), keihi (*Cinnamomi Cortex*) (*Ara & Sogawa, 2016*), and kankyo (*Zingiberis Processum Rhizoma*) reduced $PGE_2$ production (*Ara & Sogawa, 2017*). In particular, shokyo and kankyo strongly reduced LPS-induced $PGE_2$ production. These results suggested that kampo medicines that include shokyo or kankyo have anti-inflammatory effects for periodontal disease.

Both shokyo and kankyo are contained in almost all kampo medicines, and are the aqueous extracts of ginger (*Zingiber offinale* Roscoe). As described in the recent our review (*Ara et al., 2018*), shokyo is the powdered rhizome of ginger, and kankyo is the steamed and powdered rhizome of ginger. Many reports have demonstrated that ginger possesses anti-inflammatory effects as below. Ginger is clinically used as a treatment for rheumatoid arthritis, fever, emesis, nausea, and migraine headache (*Afzal et al., 2001*), and a systematic review revealed that the extracts of ginger are clinically effective as hypoanalgesic agents (*Lakhan, Ford & Tepper, 2015*). In an animal model, orally- or intraperitoneal-administrated aqueous extract of ginger reduced the serum $PGE_2$ level in rats (*Thomson et al., 2002*). Moreover, the crude hydroalcoholic extract of ginger reduced LPS-induced $PGE_2$ serum level, and improved tracheal hyperreactivity and lung inflammation in rats (*Aimbire et al., 2007*). Furthermore, ethanol extract of ginger reduced the tissue level of $PGE_2$ and improved acetic acid-induced ulcerative colitis in rats (*El-Abhar, Hammad & Gawad, 2008*).

In recent review (*Alsherbiny et al., 2019*), the effects of the aqueous extract of ginger are summarized. For example, the aqueous extract of ginger reduced ultraviolet B-induced inflammatory cytokines production in human keratinocyte HaCaT cells and mice (*Guahk et al., 2010*) and LPS-induced inflammatory cytokines in mice (*Choi et al., 2013*). Moreover, the aqueous extract of ginger has the protective effects against various organs such as liver, kidney, neuron, and heart in mice and rats. However, there are few reports on the effects on $PGE_2$ production and the arachidonic acid cascade (*Thomson et al., 2002*; *Ara & Sogawa, 2016*; *Ara & Sogawa, 2017*). Therefore, we examined the effects of shokyo and kankyo themselves on $PGE_2$ production and the arachidonic acid cascade in mouse macrophage RAW264.7 cells. We also investigated the effects of 6-shogaol at a concentration corresponding to that of these herbs.

## MATERIALS AND METHODS

### Reagents and cells

Powders of shokyo and kankyo were provided by Tsumura & Co. 3-D HPLC profiles of shokyo and kankyo were shown in Fig. S1. These powders were suspended in Dulbecco's modified Eagle's medium (D-MEM, Wako, Osaka, Japan) containing 10% heat-inactivated fetal calf serum, 100 units/ml of penicillin, and 100 mg/ml of streptomycin (culture medium), and were rotated at 4 °C overnight. Then, the suspensions were centrifuged and the supernatants were filtered through a 0.45 μm pore membrane. Lipopolysaccharide (LPS) from *Porphyromonas gingivalis* 381 was provided by Professor Nobuhiro Hanada (School of Dental Medicine, Tsurumi University, Japan). Arachidonic acid and 6-shogaol were purchased from Cayman Chemical (Ann Arbor, MI, USA). NF-κB inhibitor, BAY 11-7082, was purchased from Wako. Mitogen-activated protein kinase kinase (MAPKK/MEK)

inhibitor, PD98059, were purchased from Sigma (St. Louis, MO). Other reagents were purchased from Nacalai Tesque.

The mouse macrophage cell line RAW264.7 (RIKEN BioResource Research Center, Tsukuba, Japan) was cultured in culture medium at 37 °C in a humidified atmosphere of 5% $CO_2$.

## Measurement of cell viability

The numbers of viable cells were measured using WST-8 (Cell Counting Kit-8; Dojindo, Kumamoto, Japan) according to the manufacturer's instructions. In brief, cells were seeded onto 96-well plates (AGC Techno Glass Co., Chiba, Japan) (50,000 cells/well), and treated with shokyo or kankyo for 24 h. Then, the media were removed by aspiration and the cells were treated with a 100 µl mixture of WST-8 with culture medium for 2 h at 37 °C in a $CO_2$ incubator.

The optical density was measured (measured wavelength at 450 nm and reference wavelength at 655 nm) using an iMark microplate reader (Bio-Rad, Hercules, CA, USA), and the mean background value was subtracted from each value. Data are presented as mean $\pm$ SD ($n = 4$).

## Measurement of prostaglandin $E_2$ ($PGE_2$)

RAW264.7 cells were seeded in 96-well plates (50,000 cells/well) and incubated in culture medium at 37 °C overnight. For simultaneous treatment, cells were treated with varying concentrations of each herb in the absence or presence of LPS (100 ng/ml) for 24 h (200 µl per well) in triplicate or quadruplicate for each sample. For sequential treatment, cells were treated with medium or LPS for 30 min, and thereafter treated with medium or each herb for 24 h. After the culture supernatants were collected, viable cell numbers were measured using WST-8 as described above.

The concentrations of $PGE_2$ in the culture supernatants were measured by enzyme-linked immunosorbent assay (ELISA) according to the manufacturer's instructions (Cayman Chemical), and were adjusted by the number of viable cells. Data are presented as pg per 10,000 cells (mean $\pm$ SD).

## Measurement of cytosolic phospholipase $A_2$ ($cPLA_2$) activity

$cPLA_2$ activity was evaluated using $cPLA_2$ Assay kit (Cayman Chemical) according to the manufacturer's instructions. RAW264.7 cells were cultured in 100-mm dishes, and treated with 100 ng/ml of PgLPS for 2 h. Then, the cells were washed twice with Tris-buffered saline (TBS), transferred into microcentrifuge tubes, and centrifuged at 6,000$\times$ g for 5 min at 4 °C. Supernatants were aspirated, and cells were resuspended in TBS with 1/100 volume of protease inhibitor cocktail (Nacalai tesque) and 1/100 volume of phosphatase inhibitor cocktail (Nacalai tesque), and were homogenized with Dounce tissue grinder (Sansyo, Tokyo, Japan), Then, cells were centrifuged at 12,000$\times$ g for 15 min at 4 °C. The supernatant was collected and used as sample. To detect only $cPLA_2$ activity, samples were pretreated with 20 µM of bromoenol lactone [calcium-independent $PLA_2$ ($iPLA_2$)-specific

inhibitor, Cayman Chemical] and 20 μM of thioetheramide-PC [secretory PLA$_2$ (sPLA$_2$)-specific inhibitor, Cayman Chemical] for room temperature for 15 min. Bee venom PLA$_2$ was used as positive control.

## Measurement of cyclooxygenase (COX)-2 activity

COX-2 activity was indirectly evaluated as reported previously (*Wilborn et al., 1995*), with slight modification. In brief, to estimate COX-2 activity, RAW264.7 cells (50,000 cells/well in 96-well plate) were treated with LPS and each herb for 6 h (simultaneous treatment) or LPS for 6 h and thereafter with each herb for 1 h (sequential treatment). Then, the cells were washed and incubated in culture medium containing exogenous arachidonic acid (10 μM) for 30 min. The concentrations of PGE$_2$ in the supernatants were measured by ELISA. Data are presented as 100% at LPS alone (mean ± SD).

## Preparation of cell lysates

RAW264.7 cells were cultured in 60-mm dishes, and treated with combinations of LPS and herbs for the indicated times. Then, the cells were washed twice with TBS, transferred into microcentrifuge tubes, and centrifuged at 6,000× g for 5 min at 4 °C. Supernatants were aspirated and cells were lysed on ice in lysis buffer [50 mM Tris-HCl, pH 7.4, 1% Nonidet P-40, 0.25% sodium deoxycholate, 150 mM NaCl, one mM ethyleneglycol bis(2-aminoethylether)tetraacetic acid (EGTA), 1 mM sodium orthovanadate, 10 mM sodium fluoride, 1/100 volume of protease inhibitor cocktail, and 1/100 volume of phosphatase inhibitor cocktail] for 30 min at 4 °C. Samples were next centrifuged at 12,000× g for 15 min at 4 °C, and supernatants were collected. The protein concentration was measured using a BCA Protein Assay Reagent kit (Pierce Chemical Co., Rockford, IL, USA).

## Western blotting

The samples (50 μg of protein) were fractionated in a polyacrylamide gel under reducing conditions and transferred onto a polyvinylidene difluoride (PVDF) membrane (Hybond-P; GE Healthcare, Uppsala, Sweden). The membranes were blocked with 5% ovalbumin for 1 h at room temperature and incubated with the primary antibody for an additional 1 h. The membranes were further incubated with horseradish peroxidase-conjugated secondary antibodies for 1 h at room temperature. Protein bands were visualized with an ECL kit (GE Healthcare). Protein levels were quantified using image analysis software ImageJ (National Institutes of Health [NIH], Bethesda, MD).

Antibodies against COX-2 (sc-1745, 1:1,000 dilution), cPLA$_2$ (sc-438, 1:500 dilution), LOX-5 (LO-5, sc-515821, 1:250 dilution), annexin 1 (sc-11387, 1:1,000 dilution), actin (sc-1616, 1:1,000 dilution), NF-κB p65 (sc-372, 1:1,000 dilution), and phosphorylated p65 (Ser536) (p-NF-κB p65; sc-101752, 1:500 dilution) were purchased from Santa Cruz Biotechnology (Santa Cruz, CA). Antibodies against extracellular signal-regulated kinase (ERK; p44/42 MAP kinase antibody, 1:1,000 dilution) and phosphorylated ERK [Phospho-p44/42 MAPK (Thr202/Tyr204) (E10) monoclonal antibody, 1:2,000 dilution] were from Cell Signaling Technology (Danvers, MA). Horseradish peroxidase-conjugated anti-goat IgG (sc-2020, 1:50,000 dilution) was from Santa Cruz, and anti-rabbit IgG (1:50,000

dilution) and anti-mouse IgG (1:50,000 dilution) were from DakoCytomation (Glostrup, Denmark).

## Immunofluorescence staining

To detect p65 localization RAW264.7 cells were directly cultured on Lab-Tek chamber slides (16 well, Thermo Fisher Scientific, Waltham, MA, USA) overnight. The cells were treated with shokyo, kankyo (100 µg/ml), 6-shogaol (0, 0.1, 1, or 10 µM), or BAY 11-7082 (10 µM) for 2 h, and further treated with LPS (100 ng/ml) and shokyo, kankyo, 6-shogaol, or BAY 11-7082 for 30 min. Then, the cells were fixed with ice-cold methanol for 30 min at −20 °C. Subsequently, the cells were washed with PBS for 3 times, blocked with 1% BSA in PBS for 1 h, incubated with anti-p65 antibody (1:50 dilution) for 1 h at room temperature. After washing with PBS for three times, cells were incubated for 1 h at room temperature with Alexa Fluor488-conjugated secondary antibody (Thermo Fisher Scientific, 1:500 dilution). After washing with PBS for three times, cells were mounted with VECTASHIELD Antifade Mounting Medium with DAPI (Vector Laboratories, Burlingame, CA). Fluorescence was visualized using Axioplan2 imaging (Carl Zeiss, Inc., Oberkochen, Deutschland).

## Quantification of 6-shogaol in shokyo and kankyo

The quantification of 6-shogaol in each herb was performed by the Nagano Prefecture Pharmaceutical Association Analytical Examination Center (Nagano, Japan). In brief, one ml of samples (herbs in culture medium) was absorbed to a reverse-phase system cartridge (SepPak tC18, Waters, Milford, MA, USA). Columns were washed with two ml of 40% MeOH and one ml of 70% MeOH. Then, samples were eluted with 100% MeOH and concentrated to one ml. These samples were subjected to high-performance liquid chromatography (HPLC) with LC-20A (SHIMADZU, Kyoto, Japan). The conditions were as follows: column, X-Bridge $2.1 \times 150$ mm, three µm (Waters); solvent, 70% aqueous acetonitrile; flow rate, 0.15 min/ml; column oven, 40 °C; detection, 228 nm; and injection, 10 µl.

## Statistical analysis

Differences between the control group and experimental groups were evaluated by a two-tailed Dunnett's test. All computations were performed with the statistical program R (*R Development Core Team, 2018*). Dunnett's test was performed using the 'glht' function in the 'multcomp' package (*Hothorn, Bretz & Westfall, 2008*). The $IC_{50}$ value and its 95% confidence interval (CI) were calculated using the 'drm' function in the 'drc' package (*Ritz et al., 2015*). Values with $P < 0.05$ were considered significantly different.

# RESULTS

## Effects of shokyo and kankyo on cell viability

We first examined the effects of shokyo and kankyo on RAW264.7 cell viability. Both shokyo and kankyo reduced the cell viability in a concentration-dependent manner (Figs. 1A and 1B). However, the cell viability is hardly affected at 100 µg/ml of shokyo and kankyo, and is slightly reduced at 1,000 µg/ml. Therefore, concentrations up to 100 µg/ml of shokyo

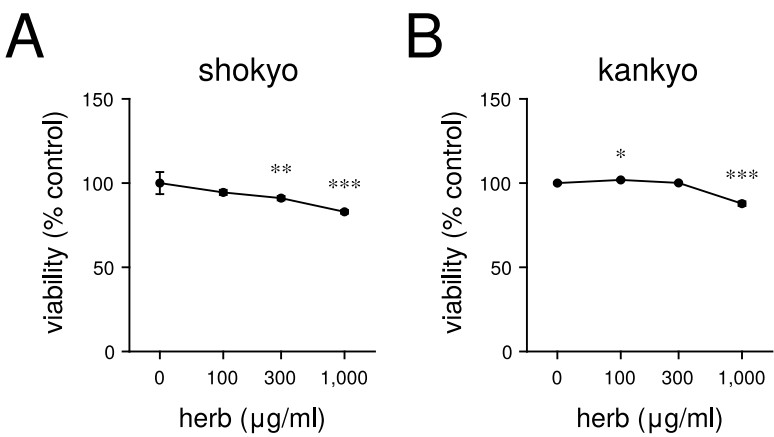

**Figure 1** **Cytotoxicity of shokyo (A) and kankyo (B).** RAW264.7 cells were treated with each herb (0, 100, 300, or 1,000 $\mu$g/ml) for 24 h. Then, the numbers of viable cells were measured by WST-8. $P$-values vs. without each herb were calculated by Dunnett's method. *$P < 0.05$, ***$P < 0.001$.

and kankyo were used in further experiments because we used the same concentration of herbs in previous studies (*Ara & Sogawa, 2016*; *Ara & Sogawa, 2017*).

## Effects of shokyo and kankyo on prostaglandin E₂ (PGE₂) production

We next examined whether shokyo and kankyo affect the production of $PGE_2$ by RAW264.7 cells. The time schedule of treatment is shown in Fig. 2A. In the simultaneous treatment experiment, RAW264.7 cells treated with 100 ng/ml of LPS produced $PGE_2$. Shokyo and kankyo (both 100 $\mu$g/ml) strongly reduced LPS-induced $PGE_2$ production (Fig. 2B).

To exclude the possibility that components in these herbs non-specifically bind the LPS receptor and inhibit LPS signaling, we performed a sequential treatment experiment. In this experiment, the cells were treated with LPS first, and the LPS receptor was not inhibited. The same results (Fig. 2C) as in the simultaneous treatment experiment were obtained, suggesting that the reduction of $PGE_2$ production is due to non-specific binding of the LPS receptor by components in shokyo and kankyo. Therefore, we performed simultaneous treatment in the following experiments.

We investigated the concentration-dependent effects of shokyo and kankyo on LPS-induced $PGE_2$ production. Both herbs reduced LPS-induced $PGE_2$ production in a concentration-dependent manner (Figs. 2D and 2E).

## Effects of kankyo and shokyo on the arachidonic acid cascade

To clarify the mechanism by which shokyo and kankyo reduce LPS-induced $PGE_2$ production more directly, we assessed the effects of these two herbs on the arachidonic acid cascade. First, we examined $cPLA_2$ activity using the homogenate of RAW264.7 cells. However, we did not detect $cPLA_2$ activity because the activity was background level (data not shown). Next, we examined the effects of shokyo and kankyo on COX activity. In order to bypass $PLA_2$, we added exogenous arachidonic acid to RAW264.7 cells treated with LPS alone or LPS plus herbs (simultaneous treatment experiment). Then, we measured the $PGE_2$ level produced by COX. Both shokyo and kankyo increased LPS-induced $PGE_2$

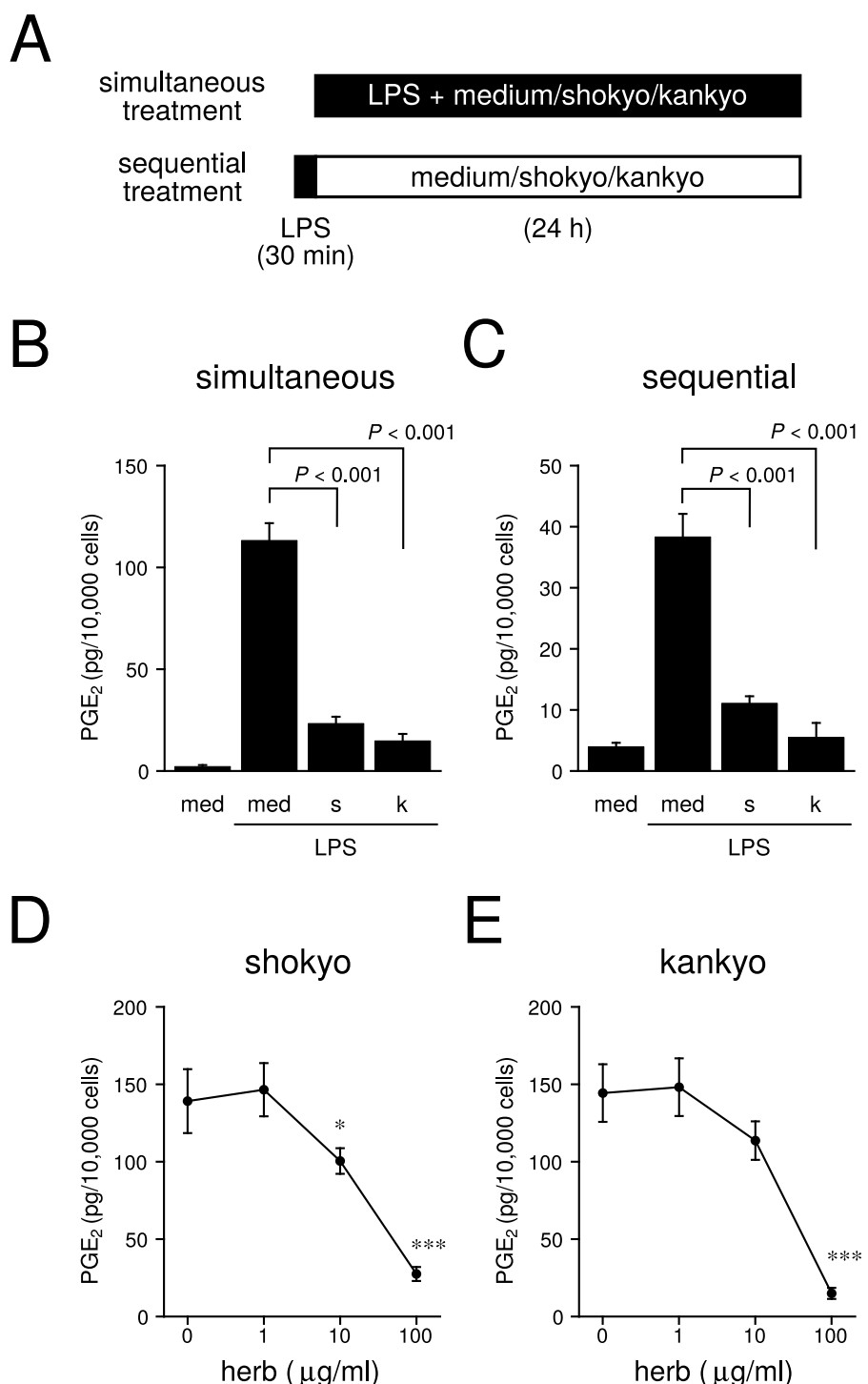

**Figure 2** **Effects of shokyo and kankyo on PGE$_2$ production.** (A) Time schedule of treatment with LPS and/or each herb. Simultaneous treatment: Cells were treated with combinations of LPS (0 or 100 ng/ml) and medium or each herb (100 μg/ml) for 24 h. Sequential treatment: Cells were treated with medium or LPS (100 ng/ml) for 30 min, washed, and further treated with medium or each herb (100 μg/ml) for 24 h. 

**Figure 2 (...continued)**
(B, C) Concentrations of PGE$_2$ were measured by ELISA, adjusted by cell number, and expressed as 100% at LPS alone (mean ± SD, $n = 3$) in simultaneous (B) and sequential (C) treatment experiments. med, medium; s, shokyo; k, kankyo. (D, E) Concentration-dependent effects of shokyo (D) and kankyo (E) on LPS-induced PGE$_2$ production. Cells were treated with combinations of LPS (100 ng/ml) and each herb (0, 1, 10, or 100 μg/ml) for 24 h. $P$-values vs. with LPS alone were calculated by Dunnett's method. $^*P < 0.05$, $^{**}P < 0.01$, $^{***}P < 0.001$.

production (Fig. 3B), suggesting that these two herbs increase COX-2 activity. Next, to exclude the effects of the change in COX-2 expression, we performed the sequential treatment experiment. As the cells were treated with LPS first in this experiment, COX-2 protein levels were considered to be comparable. Kankyo slightly increased COX-2 activity (Fig. 3C). Based on these results, shokyo and kankyo do not inhibit COX-2 activity.

Next, we examined whether shokyo and kankyo affect the expression of molecules in the arachidonic acid cascade. Shokyo and kankyo slightly reduced cPLA$_2$ expression (Fig. 3D). COX-2 was not expressed in the absence of LPS, and the treatment with LPS alone increased COX-2 expression. Shokyo and kankyo did not alter LPS-induced COX-2 expression (Fig. 3D). Moreover, shokyo and kankyo did not alter annexin 1 expression (Fig. 3D).

The expression of COX-2 is well known to be regulated by NF-κB. Therefore, we analyzed NF-κB activation by the translocation of p65, a subunit of NF-κB, into nucleus. p65 was localized at cytoplasm in untreated RAW264.7 cells. When RAW264.7 cells were treated with LPS, p65 was mainly localized at nucleus although p65 was present in cytoplasm (Fig. 4A). Shokyo and kankyo did not affect the p65 translocation into nucleus (Fig. 4A). In addition, we analyzed NF-κB activation by the level of phosphorylation of p65. Phosphorylated p65 (p-p65) was not detected without LPS treatment, and treatment with LPS alone increased the p-p65 level. Pretreatment of shokyo or kankyo for 1 h did not alter LPS-induced p65 phosphorylation (Fig. 4B), demonstrating that shokyo and kankyo did not inhibit NF-κB activity.

Moreover, we evaluated the effects of shokyo and kankyo on ERK phosphorylation. cPLA$_2$ is directly phosphorylated and activated by phosphorylated ERK (*Lin et al., 1993*; *Gijón et al., 1999*). Therefore, we examined whether shokyo and kankyo suppress LPS-induced ERK phosphorylation. LPS treatment increased ERK phosphorylation at 0.5 h and its phosphorylation was later attenuated. However, 100 μg/ml of shokyo or kankyo only slightly reduced LPS-induced ERK phosphorylation (Fig. 4C).

We also evaluated the effects of shokyo and kankyo on the lipoxygenase pathway. Lipoxygenase (LOX)-5 expression was not altered by LPS treatment. Moreover, LOX-5 expression was not affected by shokyo or kankyo (Fig. S2A). Shokyo and kankyo did not change LOX activity because LTB$_4$ production was not altered when arachidonic acid was added (Figs. S2B and S2C). Furthermore, the LTB$_4$ level was lower than that of PGE$_2$ (Figs. 3B and 3C).

## Quantification of 6-shogaol in shokyo and kankyo
6-Shogaol is one of the major and bioactive components in shokyo and kankyo. In order to assess the effects of 6-shogaol on PGE$_2$ production by RAW264.7 cells at a similar

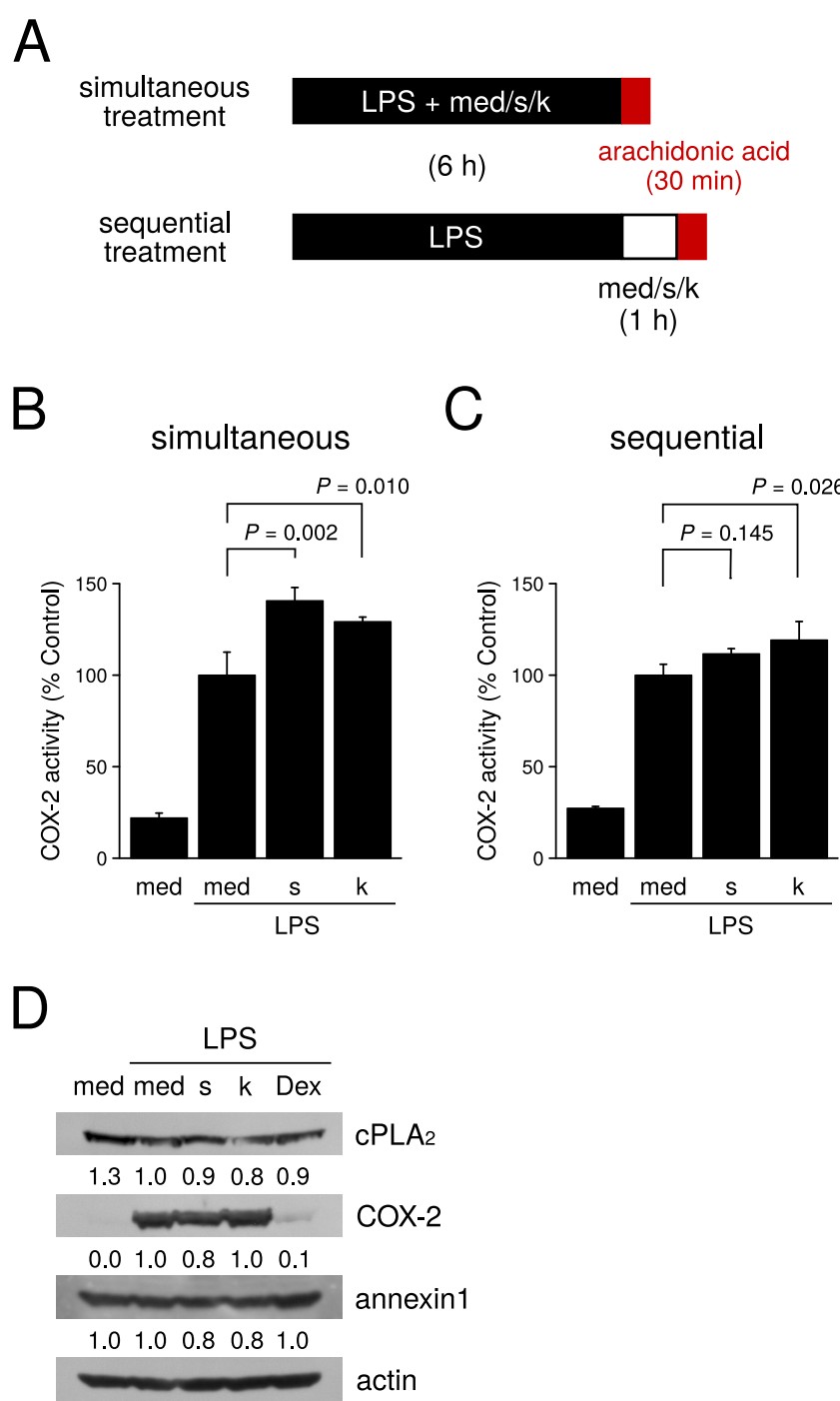

**Figure 3** **Effects of shokyo and kankyo on the arachidonic acid cascade.** (A) Time schedule of treatment with LPS and/or each herb. Simultaneous treatment: RAW264.7 cells were treated with LPS (100 ng/ml) and medium or each herb (100 μg/ml) for 6 h, washed, and then treated with 10 μM arachidonic acid for 30 min. Sequential treatment: RAW264.7 cells were treated with LPS (100 ng/ml) for 6 h, and further treated with medium or each herb (100 μg/ml) for 1 h. Then, the cells were washed and treated with 10 μM arachidonic acid for 30 min. (B, C) Effects of shokyo and kankyo on COX activity. 

**Figure 3 (…continued)**
Concentrations of PGE$_2$ were measured by ELISA, adjusted by cell number, and expressed as per 10,000 cells (mean ± SD, $n = 4$) in simultaneous (B) and sequential (C) treatment experiments. $P$-values vs. with LPS alone by Dunnett's test are indicated. (D) Effects of herbs on cPLA$_2$, annexin 1, and COX-2 expression. RAW264.7 cells were treated with a combination of LPS (0 or 100 ng/ml) and medium, each herb (100 μg/ml), or dexamethasone (100 nM) for 8 h, and protein levels were examined by Western blotting. med, medium; s, shokyo; k, kankyo; and Dex, dexamethasone. The band densities were normalized against LPS alone and actin. The values were indicated below each band.

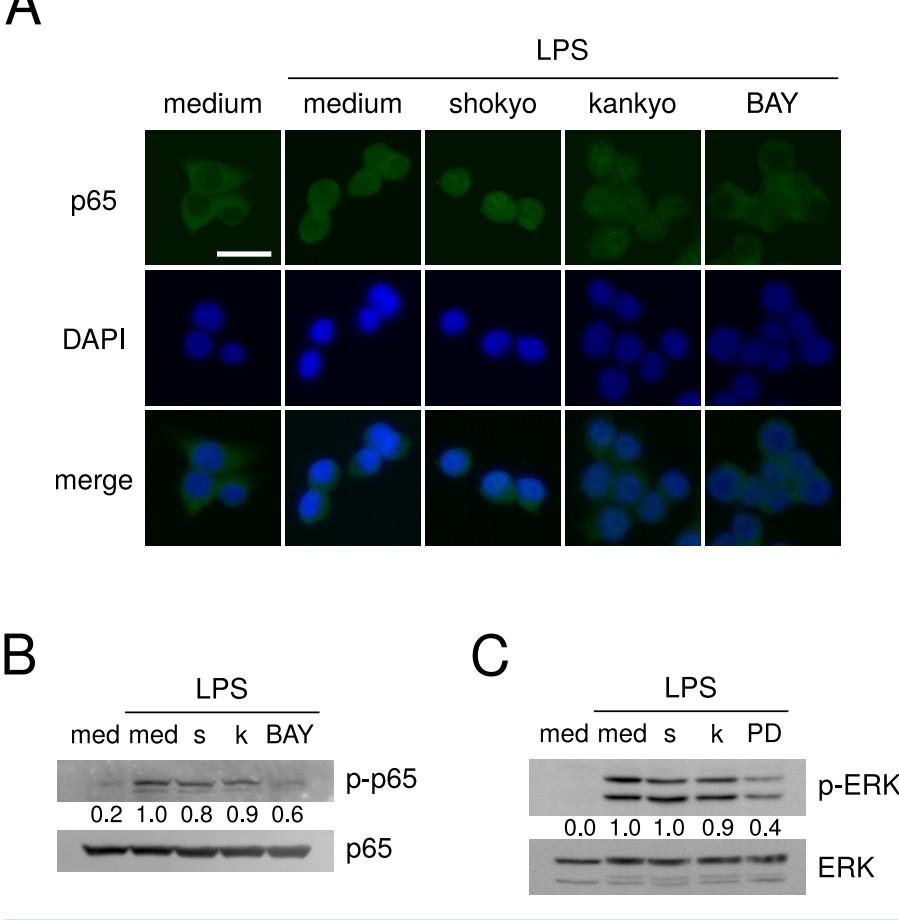

**Figure 4  Effects of shokyo and kankyo on the intracellular signal transduction.** (A) Effects of herbs on NF-κB p65 translocation into nucleus. RAW264.7 cells were treated with each herb (100 μg/ml), or BAY 11-7082 (10 μM) for 2 h, and further treated with LPS (100 ng/ml), and 6-shogaol or BAY 11-7082 for 30 min. The cellular localization of p65 was determined by immunofluorescence analysis. Nuclei of the corresponding cells were visualized with DAPI, and observed at 400× magnification. The bar represents 20 μm. (B, C) Effects of herbs on NF-κB p65 and ERK phosphorylation. RAW264.7 cells were treated with each herb (100 μg/ml), BAY 11-7082 (10 μM), or PD98059 (20 μM) for 1 h, and further treated with LPS (100 ng/ml) and each herb, BAY 11-7082, or PD98059 for 15 min. p65 and phosphorylated p65 levels (B), and ERK and phosphorylated ERK (pERK) levels (C) were examined by Western blotting. The upper band indicates ERK1 (p44 MAPK) and the lower band indicates ERK2 (p42 MAPK) in (C). med, medium; s, shokyo; k, kankyo; BAY, BAY 11-7082, and PD, PD98059. The band densities were normalized against LPS alone, and p65 or ERK. The values were indicated below each band.

**Table 1  Concentrations of 6-shogaol in shokyo and kankyo by HPLC analysis.**

| | Concentration of herb | |
| | 5 mg/ml | 100 μg/ml |
| --- | --- | --- |
| shokyo | 2.97 μM | 59.4 nM |
| kankyo | 4.87 μM | 97.4 nM |

**Notes.**
Chromatography profiles are shown in Fig. S3.

concentration to shokyo and kankyo, we quantified the amount of 6-shogaol in shokyo and kankyo. HPLC analysis revealed that the 5 mg/ml shokyo and kankyo solutions used in this study contained 2.97 μM and 4.87 μM 6-shogaol, respectively (Table 1 and Fig. S3). Thus, 100 μg/ml of shokyo and kankyo contained 59.4 nM and 97.4 nM 6-shogaol, respectively (Table 1).

## Effects of 6-shogaol on PGE$_2$ production and molecular expression in the arachidonic acid cascade

We investigated the effects of 6-shogaol on LPS-induced PGE$_2$ production by RAW264.7 cells. 6-Shogaol reduced LPS-induced PGE$_2$ production in a concentration-dependent manner (Fig. 4A). One hundred nM 6-shogaol reduced PGE$_2$ production to approximately 50%, and 1,000 nM (= 1 μM) 6-shogaol inhibited PGE$_2$ production. The IC$_{50}$ value of 6-shogaol was 105 nM (95% CI: 27.7–182 nM). However, comparing with the calculated 6-shogaol concentration in shokyo and kankyo based on the results in Table 1, the effects of 6-shogaol on PGE$_2$ production were weaker than those of shokyo and kankyo (red and blue lines in Fig. 5A, respectively).

The effects of 6-shogaol on the expression of molecules in the arachidonic acid cascade and intracellular signal transduction pathways were evaluated, but 10 μM 6-shogaol did not affect the expression of cPLA$_2$, annexin 1, or COX-2 (Fig. 5B). Moreover, 6-shogaol did not affect LPS-induced p65 translocation into nucleus (Fig. 6A), or p65 phosphorylation (Fig. 6B). Furthermore, 6-shogaol did not alter ERK phosphorylation (Fig. 6C).

## DISCUSSION

There are few reports on the effects of shokyo or kankyo, which are aqueous extracts of ginger, on the arachidonic acid cascade. We previously examined the effects of shokyo and kankyo on the arachidonic acid cascade in HGFs, and suggested that these herbs inhibit cPLA$_2$ activity because they did not inhibit COX-2 activity or suppress cPLA$_2$ and COX-2 expression (*Ara & Sogawa, 2016*; *Ara & Sogawa, 2017*). In this study, we examined the effects of these herbs in macrophage-like RAW264.7 cells and obtained similar results (Figs. 3B–3D). In addition, shokyo and kankyo did not alter annexin 1 (also named lipocortin1) expression (Fig. 3D), which is produced by glucocorticoids and inhibits cPLA$_2$ activity (*Gupta et al., 1984*; *Wallner et al., 1986*). Moreover, shokyo and kankyo did not alter NF-κB p65 translocation into nucleus (Fig. 4A), p65 phosphorylation (Fig. 4B), or ERK phosphorylation (Fig. 4C). Because NF-κB activation is required to induce COX-2 expression, our results that shokyo and kankyo did not alter COX-2 expression (Fig. 3D)

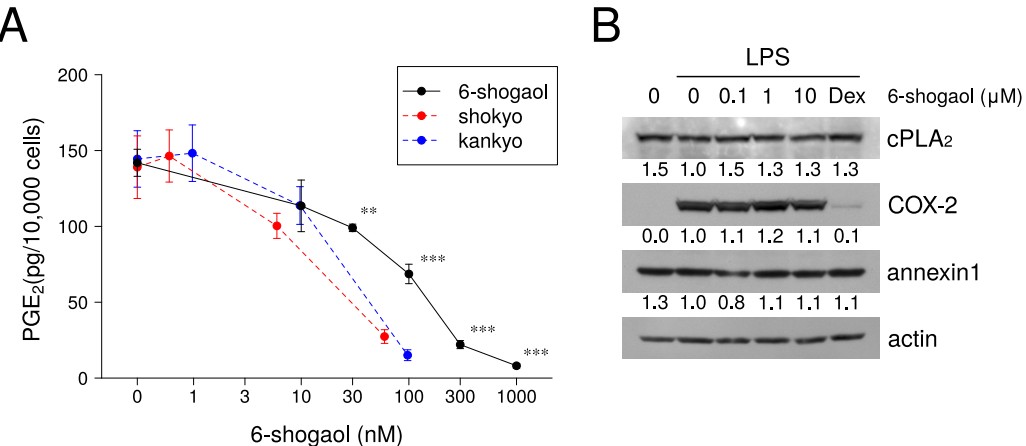

**Figure 5** **Effects of 6-shogaol on LPS-induced PGE$_2$ production, and the arachidonic acid cascade.**
(A) Effects of 6-shogaol on LPS-induced PGE$_2$. RAW264.7 cells were treated with LPS (100 ng/ml) and 6-shogaol (0 to 1,000 nM) for 24 h. Concentrations of PGE$_2$ were measured by ELISA, adjusted by cell number, and expressed as per 10,000 cells (mean $\pm$ SD, $n = 3$). $P$-values vs. with LPS alone were calculated by Dunnett's method. $*P < 0.05$, $**P < 0.01$, $***P < 0.001$. Red and blue lines represent the effects of shokyo and kankyo in Figs. 2D and 2E, respectively. The concentrations of 6-shogaol in shokyo and kankyo were calculated using the results shown in Table 1. (B) Effects of 6-shogaol on cPLA$_2$, annexin 1, and COX-2 expression. RAW264.7 cells were treated with a combination of LPS (0 or 100 ng/ml) and 6-shogaol (0, 0.1, 1, or 10 μM), or dexamethasone (100 nM) for 8 h, and protein levels were examined by Western blotting. The band densities were normalized against LPS alone and actin. The values were indicated below each band.

are consistent with those in NF-κB activation. In addition, because ERK phosphorylation is required to activate cPLA$_2$, our results suggest that shokyo and kankyo did not alter cPLA$_2$ activation. Unfortunately, we could not directly evaluate the effects of shokyo, kankyo, and 6-shogaol on cPLA$_2$ activity in this study, because cPLA$_2$ activity of RAW264.7 cells were not detected. However, these results suggest that shokyo and kankyo inhibit cPLA$_2$ activity in RAW264.7 cells and in HGFs, and their effects may be cell type-nonspecific.

As a possible mechanism by which shokyo and kankyo reduced LPS-induced PGE$_2$ production, components of shokyo and kankyo may bind to LPS receptors on the cell surface and inhibit LPS signaling. However, even after the removal of LPS, shokyo and kankyo reduced LPS-induced PGE$_2$ production in the sequential treatment experiment (Fig. 2C). If some components in these herbs either competitively or noncompetitively block LPS receptors or reduce PGE$_2$ production, LPS-induced NF-κB p65 and ERK phosphorylation should have been inhibited. However, shokyo and kankyo did not suppress LPS-induced NF-κB and ERK phosphorylation (Figs. 4B and 4C). These results therefore excluded this hypothesis, and the target sites of shokyo and kankyo are present intracellularly.

Gingerols and shogaols are the major components of ginger (reviewed in *Ara et al., 2018*); therefore, shokyo and kankyo contain these components. With prolonged storage or heating of ginger, gingerols are dehydrated and converted to shogaols (*Afzal et al., 2001*). Because kankyo is subjected to heat processing, kankyo contains a larger amount of

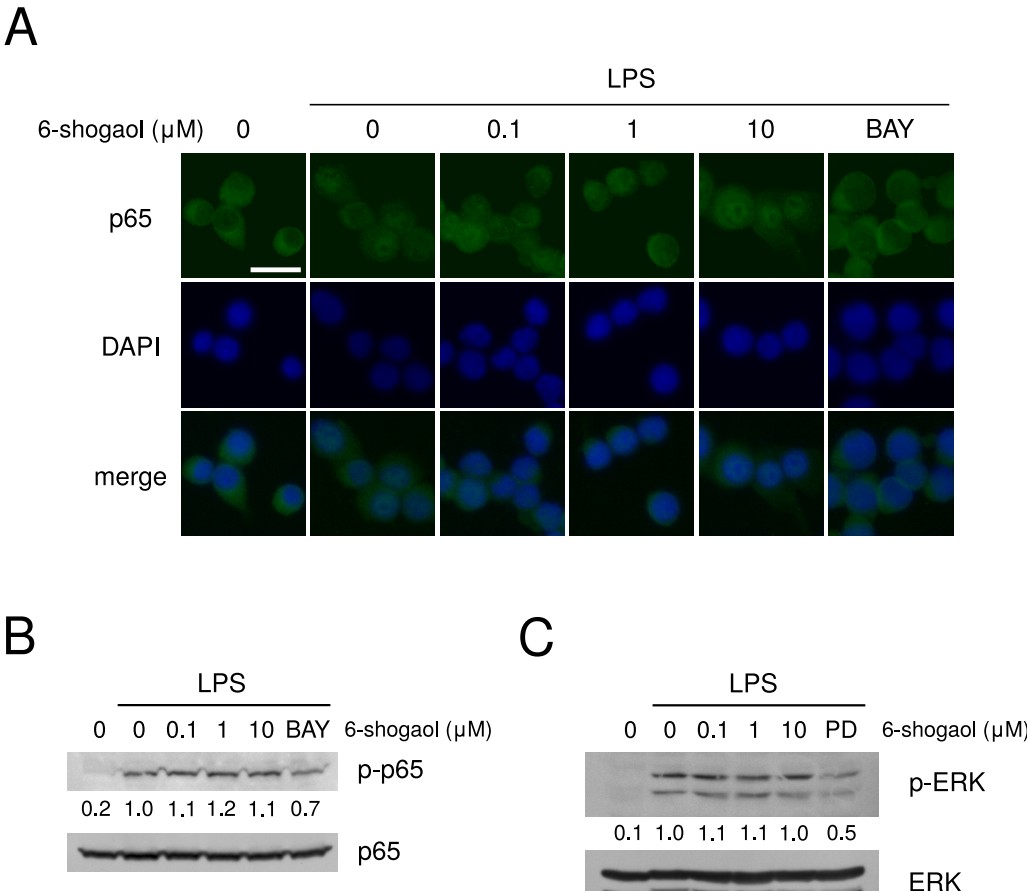

**Figure 6** **Effects of 6-shogaol on the intracellular signal transduction.** (A) Effects of 6-shogaol on NF-κB p65 translocation into nucleus. RAW264.7 cells were treated with 6-shogaol (0, 0.1, 1, or 10 μM), or BAY 11-7082 (10 μM) for 2 h, and further treated with LPS (100 ng/ml), and 6-shogaol or BAY 11-7082 for 30 min. The cellular localization of p65 was determined by immunofluorescence analysis. Nuclei of the corresponding cells were visualized with DAPI, and observed at 400× magnification. The bar represents 20 μm. (B, C) Effects of 6-shogaol on NF-κB p65 and ERK phosphorylation. RAW264.7 cells were treated with 6-shogaol (0, 0.1, 1, or 10 μM), BAY 11-7082 (10 μM), or PD98059 (20 μM) for 1 h and further treated with LPS (100 ng/ml), and 6-shogaol, BAY 11-7082, or PD98059 for 15 min. p65 and phosphorylated p65 levels (B), and ERK and phosphorylated ERK (pERK) levels (C) were examined by Western blotting. BAY: BAY 11-7082, and PD: PD98059. The band densities were normalized against LPS alone, and p65 or ERK. The values were indicated below each band.

shogaols than shokyo as shown in Fig. S1. Among them, 6-shogaol is one of the bioactive components, and was reported to reduce PGE$_2$ production (*Ara et al., 2018*). Indeed, kankyo contained 1.7-times the amount of 6-shogaol as shokyo (approximately 60 nM and 100 nM 6-shogaol in 100 μg/ml of shokyo and kankyo, respectively). However, because shokyo and kankyo are water-extracts of ginger, the amount and effects of gingerols and shogaols are thought to be lower than those of organic solvent-extracts such as methanol.

Next, we will discuss the effects of 6-shogaol on PGE$_2$ production. In this study, the IC$_{50}$ value of 6-shogaol for PGE$_2$ production was approximately 100 nM (Fig. 5A). This IC$_{50}$

value is consistent with that in previous reports: approximately 100 nM in IL-1$\beta$-treated human oral keratinocytes (*Kono et al., 2014*) and approximately 60 $\mu$g/ml (= 217 nM) in LPS-treated U937 cells (*Lantz et al., 2007*). However, the IC$_{50}$ value in this study was considered to be insufficient to inhibit the arachidonic acid cascade, as described below. The IC$_{50}$ value of 6-shogaol for COX-2 activity is 2.1 $\mu$M in A549 cells (*Tjendraputra et al., 2001*). In another report, 6-shogaol did not inhibit COX-activity in a cell-free experimental model (*Van Breemen, Tao & Li, 2011*). Although we did not examine the effects of 6-shogaol on COX-2 activity, 100 nM 6-shogaol was considered to not affect COX-2 activity because 100 $\mu$g/ml of kankyo, which contains approximately 100 nM 6-shogaol, did not inhibit COX-2 activity. Moreover, 0.17 $\mu$M 6-shogaol reduced COX-2 activity to approximately 70% in IL-1$\beta$-treated human oral keratinocytes (*Kono et al., 2014*). The reported concentrations of 6-shogaol required for the reduction of COX-2 expression were higher than that in our study. The expression of COX-2 mRNA was reduced to approximately 70% by 0.17 $\mu$M 6-shogaol (*Kono et al., 2014*). However, the expression of COX-2 protein was not affected by 1 $\mu$M 6-shogaol, was slightly reduced by 5 $\mu$M, and was significantly reduced by 10 $\mu$M in LPS-treated mouse microglial BV-2 cells (*Ha et al., 2012*). Therefore, our results that shokyo and kankyo did not inhibit COX activity are consistent with these previous reports. Similar results were observed in mouse skin (*Kim et al., 2005*). Similarly, a high concentration of 6-shogaol was reported to be required for the inhibition of NF-$\kappa$B activation. PMA-induced NF-$\kappa$B promoter activity was reduced to approximately 50%, but p65 phosphorylation was not affected by 5 $\mu$M shogaol in human breast carcinoma cells (*Ling et al., 2010*). In this study, even 10 $\mu$M 6-shogaol did not affect cPLA$_2$ or COX-2 expression, p65 translocation to nucleus (Fig. 6A), or p65 phosphorylation (Fig. 6B) Therefore, our results are consistent with these previous results. Our results suggested that 6-shogaol reduces LPS-induced PGE$_2$ production via the inhibition of cPLA$_2$ activity because the remaining and probable target site in the arachidonic acid cascade is cPLA$_2$.

However, the effects of shokyo and kankyo cannot be accounted for by only 6-shogaol. Although 100 $\mu$g/ml of kankyo reduced LPS-induced PGE$_2$ production to approximately 10% (Fig. 2E), 100 nM 6-shogaol reduced it to approximately 50% (Fig. 5A). Therefore, some components other than 6-shogaol may be involved in the reduction of PGE$_2$ production. It was previously reported that 6-gingerol is the most abundant in hangeshashinto, which contains kankyo (*Kono et al., 2014*). Moreover, the amount of 6-shogaol is approximately half of that of 6-gingerol, and the amounts of 8-gingerol, 10-gingerol, 8-shogaol, and 10-shogaol are smaller (*Kono et al., 2014*). 6-Shogaol reduced PGE$_2$ production the most, followed by 6-gingerol, 8- and 10-gingerol, and slight reduction by 8- and 10-shogaol (*Kono et al., 2014*). Thus, the effects of 6-shogaol on the reduction of PGE$_2$ production are the strongest and those of other components are weak. Therefore, gingerols and shogaols may have additive effects on PGE$_2$ production. Next, we will discuss the effects of gingerols and shogaols on the arachidonic acid cascade. The IC$_{50}$ values of these components for the inhibition of COX-2 activity are on the order of $\mu$M, similar to 6-shogaol (*Tjendraputra et al., 2001*; *Van Breemen, Tao & Li, 2011*). Moreover, 6-, 8-, and 10-gingerol reduced COX-2 expression at the $\mu$M order (*Lantz et al., 2007*). Among

ginger extracts, 10 $\mu$M 10-gingerol, 6-shogaol, 8-shogaol, and 10-shogaol reduced cPLA$_2$ activity to approximately 50% (*Nievergelt et al., 2011*). Although the concentrations of these components in shokyo and kankyo are lower than 10 $\mu$M, shokyo and kankyo may inhibit cPLA$_2$ activity by their additive effects.

We next evaluated the effects of shokyo and kankyo on the LOX pathway. However, RAW264.7 cells produced only a small amount of LTB$_4$ regardless of the presence of LPS. Moreover, shokyo and kankyo did not affect LOX-5 expression or LOX activity, suggesting that the LOX pathway is not active in macrophages, and that shokyo and kankyo do not inhibit the lipoxygenase pathway. Aspirin-induced asthma (AIA) is induced by the ingestion of acid nonsteroidal anti-inflammatory drugs (NSAIDs), and is considered to be caused by leukotrienes, which are increased by acid NSAIDs an contract the bronchus (*Vaszar & Stevenson, 2001*; *Bochenek et al., 2002*). Similarly, acid NSAIDs exacerbate general asthma. Based on our findings that (1) shokyo and kankyo inhibit upstream of the arachidonic acid cascade, (2) they did not inhibit the cyclooxygenase pathway, and (3) they did not enhance lipoxygenase pathways, shokyo and kankyo may reduce leukotriene production. Therefore, shokyo and kankyo may not exacerbate asthma, including AIA. As such, shokyo and kankyo may be safely used for patients with asthma, including AIA, instead of conventional anti-inflammatory drugs. Moreover, as oral-administrated ginger protected aspirin-induced gastric ulcers in rats (*Wang et al., 2011*), shokyo and kankyo may be available as anti-inflammatory drugs instead of NSAIDs.

Next, we will discuss about protective effect on gastric ulcer. In general, ginger reduced PGE$_2$ production in macrophages and inflammatory site. In contrast, orally administered cuttlebone complex (CBC), which includes fresh ginger roots, demonstrated a protective potentiality against indomethacin-induced gastric ulcer in rats via increment of the indomethacin-declined PGE$_2$ levels in the stomach (*Chien et al., 2015*). Therefore, ginger has opposite effects between inflammatory cells and gastric mucosal cells. Moreover, ginger powder protected aspirin-induced gastric ulcers in rat (*Wang et al., 2011*). These results suggest that ginger has both anti-inflammatory effect and protective effect on gastric ulcers in contrast to acid NSAIDs.

There are several limitations in this study. We evaluated only cPLA$_2$ among PLA$_2$ isotypes. Therefore, in the future, the evaluation of the effects of shokyo and kankyo on iPLA$_2$ and sPLA$_2$—expressions and activities—will be needed. Moreover, the evaluation of the anti-inflammatory effects of shokyo and kankyo on the inflammatory diseases such as periodontal disease or stomatitis by animal models will be needed.

## CONCLUSION

We demonstrated that shokyo and kankyo, which are water-extracted fractions of ginger, reduced LPS-induced PGE$_2$ production in RAW264.7 cells. Their mechanisms of action were suggested to via the inhibition of cPLA$_2$ activity because both herbs did not inhibit COX activity or suppress the expression of molecules in the arachidonic acid cascade.

## ACKNOWLEDGEMENTS

We thank Prof. Nobuyuki Udagawa (Department of Biochemistry, Matsumoto Dental University) and Prof. Naoyuki Takahashi (Institute for Oral Science, Matsumoto Dental University) for their advice on our study.

### Funding

The study was supported by funding from JSPS KAKENHI Grant Number JP16H05144, the Nagano Society for the Promotion of Science, and a Scientific Research Special Grant from Matsumoto Dental University. The funders had no role in study design, data collection and analysis, decision to publish, or preparation of the manuscript.

### Grant Disclosures

The following grant information was disclosed by the authors:
JSPS KAKENHI: JP16H05144.
Nagano Society for the Promotion of Science.
Scientific Research Special Grant from Matsumoto Dental University.

### Competing Interests

The authors declare there are no competing interests.

### Author Contributions

- Toshiaki Ara conceived and designed the experiments, performed the experiments, analyzed the data, contributed reagents/materials/analysis tools, prepared figures and/or tables, authored or reviewed drafts of the paper, approved the final draft.
- Masanori Koide performed the experiments, analyzed the data, contributed reagents/materials/analysis tools, prepared figures and/or tables, approved the final draft.
- Hiroyuki Kitamura performed the experiments, contributed reagents/materials/analysis tools, prepared figures and/or tables, approved the final draft.
- Norio Sogawa conceived and designed the experiments, analyzed the data, authored or reviewed drafts of the paper, approved the final draft.

### Data Availability

The raw data are available as Supplemental Files.

### Supplemental Information

Supplemental information for this article can be found online at http://dx.doi.org/10.7717/peerj.7725#supplemental-information.

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
