# Peer review of "Effects of shokyo (Zingiberis Rhizoma) and kankyo (Zingiberis Processum Rhizoma) on prostaglandin E2 production in lipopolysaccharide-treated mouse macrophage RAW264.7 cells"

_PeerJ, doi:10.7717/peerj.7725_

## Round 0.1 · original submission · Major Revisions

As you will see, our reviewers found your manuscript valuable. There are, nonetheless, many instances in your paper where the description in the text does not agree with the schemes (e.g. Fig.3A disagrees with lines 96-97, etc.etc.etc.) units are in error (e.g. Table 1, in the 100ug/ml column units should by nM instead of uM) and so forth. Please address all of those inconsistencies.

I agree with reviewer #2 regarding the absence of evidence that the effect of shokyo or kankyo relies on inhibition of PLA2.

Reviewer 1 ·

Basic reporting

This manuscript dose an excellent job demonstrating significant effect of Shokyo and Kankyo on prostaglandin E2 production. The article is well written, treats an actual problem. Also, it has focus on importance of traditional Asian medicine.

Experimental design

The experiment has designed very well, with appropriate statistical analysis.

Validity of the findings

The findings are valid and interesting and it has focused on role of traditional Japanese medicine and also spread organic and easier way for more usage of traditional anti-inflammatory drugs.
Japanese traditional medicine uses most of the Chinese therapies including acupuncture and moxibustion, but Kampō in its present-day sense is primarily concerned with the study of herbs. Kambo formulas were introduced from China and it is fully integrated into the modern health care system in Japan. Kampo can be an ideal choice of therapy.

Additional comments

I think this article excellent and is an important link between Shokyo and Kankyo and their water-extracted fractions of ginger with mechanisms of action via the inhibition of cPLA2 activity. This is an interesting topic of research. I personally enjoyed the role of gingerols and shogaols in the reduction of LPS-induced PGE2 production. And a new significant role of these extracts for inflammatory diseases.

Reviewer 2 ·

Basic reporting

Some specific questions are as follows:
1. Lines 96 and 97: The method of measuring cyclooxygenase (COX)-2 activity is inconsistent with Figure 3A.
2.Line 160: Authors claimed that Shokyo and kankyo (both 100 µg/ml) strongly reduced LPS-induced PGE2 production (Figure 2B), and the effects of kankyo were stronger than those of kankyo. The expression is ambiguous and inconsistent with it in the section “ABSTRACT”.
3. The dosage of 6-shogaol (Lines 215-223) is inconsistent with Figure 4A. There are some confusion and mistakes of its units.
4. Some experimental results are delineated vaguely. For example, the descriptions of cell viability on line 151, Western Blot analysis of ERK phosphorylation in “RESULTS” (lines 215 to 223) and in “DISCUSSION” (line 239) are all confusing.

Experimental design

5. It was absent of direct evidences to prove that shokyo and kankyo inhibited cPLA 2 activity. The conclusion about COX-2 activity (lines 174 to 184), one of the main point of this article, was obtained through an indirect experiment. It is better to have more direct experimental evidence here, for example, enzyme activity testing.
6. In addition, the Western blot results (Figures 3 and 4) showed that neither the herbs extraction nor 6-shogaol altered LPS-induced expression of COX-2, cPLA2 or annexin 1, and there was also no significant inhibitory activity on LPS-induced phosphorylation of NF-κB p65 and ERK. The experiment lacked corresponding positive control. Besides, the relative gray value should be calculated.
7. It is not enough just to evaluate p65 phosphorylation level for researching the role of shokyo and kankyo on NF-κB signal pathway, and some complementary experiments will make the results more convincing, for example, some detection about the phosphorylation of I-κB or p65 transportation to the nucleus.

Validity of the findings

no comment

Additional comments

The literature and background introduction are quite sufficient.
There are many obvious errors, confusing statements in this manuscript and the experimental design needs to be improved.

Annotated reviews are not available for download in order to protect the identity of reviewers who chose to remain anonymous.

·

Basic reporting

The manuscript "Effects of shokyo (Zingiberis Rhizoma) and kankyo (Zingiberis Processum Rhizoma) on prostaglandin E2 production in lipopolysaccharide-treated mouse macrophage RAW264.7 cells” was professionally represented in unambiguous clear English language throughout the manuscript with structure aligned with PerrJ standard.
Additionally, raw data have been supplied with high quality relevant figures. The introduction elaborated the context with a need for further consideration of certain studies and reviews of aqueous ginger and its effect of PGE2 levels in-vitro and in-vivo which will be thoroughly mentioned later on.

Experimental design

Toshiaki Ara et al represented a manuscript aligned with the PeerJ scope with consistently defined research questions. All the methodologies are well described and the data is analysed in a satisfactory modern manner.

Validity of the findings

Robust data with well discussed findings. Nevertheless, The metabolomic profiling of shokyo and kankyo materials is to be provided if not declared in the certificated of analysis provided by Tsumura & Co and the COA provided by the commercial vendor is to be added to the supplementary section.

Additional comments

A) The anti-inflammatory mechanisms of the ginger armoury of phytochemicals and either aqueous or alcoholic extracts were comprehensively vetted in the recent review (Ameliorative and protective effects of ginger and its main constituents against natural, chemical and radiation-induced toxicities: A comprehensive review, Food and Chemical Toxicology, 2018 https://doi.org/10.1016/j.fct.2018.10.048) for both the in-vitro and the in-vivo studies. This review should have been considered and utilised in the introductory section to draw the need for more studies on the aqueous ginger extract to be discussed briefly. Furthermore, some studies elaborated in the aforementioned review are to be considered in the discussion and/or the introduction sections for examples:
- The inhibition of induced cytokines in HaCaT cells and Mice by aqueous ginger extract (Zingiber officinale protects HaCaT cells and C57BL/6 mice from ultraviolet B-induced inflammation J. Med. Food, 13 (3) (2010), pp. 673-680).
- The declined expression of IFN-γ, IL-6 and iNOS with inhibition of the NF-k B activation in LPS challenged mice by aqueous ginger extract (Y.Y. Choi, M.H. Kim, J. Hong, S.-H. Kim, W.M. YangDried ginger (zingiber officinalis) inhibits inflammation in a lipopolysaccharide-induced mouse model Evid. Based Complement Alternat. Med., 2013 (2013), p. 914563).
- The orally administered Cuttlebone complex (CBC) including fresh ginger roots demonstrated a protective potentiality against indomethathin-induced gastric ulcer in rats via reducing the gastric ulcerous lesions and significant increment of the indomethacin-declined PGE2 levels in the stomach in a dose dependent manner (M.-Y. Chien, Y.-T. Lin, F.-C. Peng, H.-J. Lee, J.-M. Chang, C.-M.Yang, C.-H. Chen, Gastroprotective potential against indomethacin and safety assessment of the homology of medicine and food formula cuttlebone complex Food Funct., 6 (8) (2015), pp. 2803-2812).

B) Discuss the ginger up-regulatory effect on PGE2 in the stomach and its down-regulatory effect in macrophages and inflammation site.

C) Global metabolomic profiling of shokyo and kankyo used materials for robustness and repeatability and QC purposes in any further studies.
D) A section on the limitations of the study and ideas for future research could be included.

---

## Round 0.2 · Minor Revisions

Please clarify the remaining issues

Reviewer 2 ·

Basic reporting

The authors of this manuscript have carefully answered the questions raised in the previous review. Some experiments were repeated or supplemented, and several obvious errors in the writing and data were corrected. However, there are still a few minor issues in this article that need to be noted and modified.

Experimental design

no comment

Validity of the findings

1. The author does not have a good understanding of the Questions 1. They tried to convert the amount of PGE2 to percent (%), and indicate as COX-2 activities (Figures 2B and 2C). Obviously, this is not appropriate. The problem with this part is the inconsistency between figures and texts. The author claimed in MATERIALS AND METHODS that the cells were treated with LPS and each herb for 6 h (simultaneous treatment) or LPS for 6 h and thereafter with each herb for 1 h (sequential treatment). Then, exogenous arachidonic acid (10 µM) for 30 min (Lines 118-121). This is consistent with the experimental method described in the annotation(Lines 507-511), But it is completely inconsistent with the “Time schedule of treatment” shown in Figure 3A.
2. I still don't see any image results of Western blots were quantified by densitometry using image software. Of course, this is just a suggestion.
3. A large number of WB tests were repeated and positive controls were added. But the stripes of WB should not be spliced( in Figure 6C).

Additional comments

no comment

---

## Author Rebuttal · Round 0.2

August 2, 2019
Dear Pedro Silva, Academic Editor:

Thank you for your e-mail of May 11, with regard to our manuscript (Article ID: 36484) entitled "Effects of shokyo (*Zingiberis Rhizoma*) and kankyo (*Zingiberis Processum Rhizoma*) on prostaglandin $E_2$ production in lipopolysaccharide-treated mouse macrophage RAW264.7 cells" by Toshiaki Ara *et al*. Moreover, we would express our thanks to you again and reviewers for reviewing this manuscript and helpful comments. I am sending therewith our revised manuscript with changes indicated by using red color and figures, with sheet detailing our response to the points raised and the changes we have made.

I hope you will find the data interesting and would consider whether this manuscript is acceptable for publication in PeerJ.

Thank you for your kind consideration.

Sincerely yours,

Toshiaki Ara, D.D.S., Ph.D.

Department of Pharmacology

Matsumoto Dental University

1780 Gobara Hirooka, Shiojiri, Nagano 399-0781, Japan

Phone: 81-263-51-2103

FAX: 81-263-51-2102

E-mail: toshiaki.ara@mdu.ac.jp

Comments to Editor

We would express our thanks to the editor for helpful comments.

1. As the reviewer2 pointed out, the concentrations of 6-shogaol in Table 1 and original Figure 4 were wrong. Therefore, we corrected the unit of concentration properly (μM to nM) in Table 1 and Figure 5 (in revised version).

2. We evaluated the inhibitory effect of shokyo and kankyo on COX-2 using the *in vivo* method by Wilborn *et al.* (1995) in which arachidonic acid is added. In the first submission, we denoted the activity as the amount of $PGE_2$ production, but this representation was difficult to understand. Therefore, we converted the amount of $PGE_2$ to percent (%), and indicated as COX-2 activities.

3. As the reviewer2 pointed out, we did not provide the direct evidences about the inhibitory effects of shokyo, kankyo, and 6-shogaol on $cPLA_2$ activity. In fact, we tried to evaluate $cPLA_2$ activity in several conditions using the commercial product ($cPLA_2$ Assay kit, Cayman Chemical). However, unfortunately we could not detect $cPLA_2$ activity in RAW264.7 cells, and therefore, we could not evaluate the inhibitory effects of these components on $cPLA_2$ activity. For these reasons, we concluded that shokyo and kankyo may inhibit $cPLA_2$ activity although the methods we used in this study are indirect.

Answers to comments of Reviewer 1

We would express our thanks to the reviewer for reviewing this manuscript and helpful comments.

Answers to comments of Reviewer 2

We would express our thanks to the reviewer for reviewing this manuscript and helpful comments.

1) As the reviewer pointed out, we used the indirect experimental method to evaluate COX-2 activity, and therefore the method is inconsistent with Figure 3. We described the reason that we used this indirect experimental method in answer 5). However, to clarify "the inhibitory effect on COX-2 activity", we converted the amount of $PGE_2$ to percent (%), and indicated as COX-2 activities **(Figures 2B and 2C)**.

2) As the reviewer pointed out, the expression that "the effect of kankyo were stronger than those of kankyo" is wrong. Therefore, we deleted the concerning sentence in Abstract and Results sections.

3) As the reviewer pointed out, the concentrations of 6-shogaol in Table 1 and Figure 4 were wrong. Therefore, we corrected the unit of concentration properly (μM to nM).

4) We corrected the vague sentences of results that the review pointed out **(Page 5, lines 187–188, Page 6, lines 259–261)**.

5) As the reviewer pointed out, we did not provide the direct evidences that shokyo, kankyo, and 6-shogaol inhibit $cPLA_2$ and COX-2 activity. The reasons that we did/could not evaluate $cPLA_2$/COX-2 activities are described below.

- In fact, we tried to evaluate $cPLA_2$ activity using the commercial product ($cPLA_2$ Assay kit, Cayman Chemical) in several conditions. However, we could not detect $cPLA_2$ activity in RAW264.7 cells, and therefore, we could not evaluate the inhibitory effects of these components on $cPLA_2$ activity. For these reasons, although the methods we used in this study are indirect, we concluded that shokyo and kankyo may inhibit $cPLA_2$ activity. We added the sentences concerning this result that we could not detect $cPLA_2$ activity in Materials and Methods **(Page 3, lines 103–115)**, Results **(Page 5, lines 209–210)**, and Discussion **(Page 7, lines 281–283)**.

- In addition, we have used the commercial product (COX Inhibitor Screening Assay Kit, Cayman Chemical) to evaluate the effect of several kampo medicines on COX-2 activity. However, we could not obtain the adequate inhibitory effects — perhaps due to the short reaction time in this method. Therefore, to evaluate inhibitory effect of shokyo and kankyo on COX-2, we used the *in vivo* method by Wilborn *et al.* (1995) in which arachidonic acid is added. In fact, the phrase "Determination of COX activity" is present in Materials and Methods section of this literature.

6) As the reviewer pointed out, we included positive controls (dexamethasone for COX-2 and annexin1, BAY 11-7082 for p65, and PD98059 for ERK) in Western blotting analysis. Moreover, we quantified the relative gray values of bands and indicated the values below each band **(Figures 3–6)**. We added the sentence of this methods in Materials and Methods **(Page 2, lines 77–79, Page 4, lines 141–142)** and Figure legend **(Figures 3–6)**.

7) As the reviewer pointed out, we added the data concerning p65 translocation to nucleus by immunohistochemistry to confirm the effect of shokyo, kankyo, and 6-shogaol on NF-$\kappa$B activity. We observed the results that shokyo, kankyo, and 6-shogaol did not affect LPS-induced p65 translocation to nucleus. We add these results to Abstract **(Page 1, line 23)**, Materials and Methods **(Page 4, lines 153–165)**, Results **(Page 6, lines 226–230 and 264-266)**, and Discussion **(Page 7, lines 276–281, Page 8, lines 327–328)**.

Answers to comments of Reviewer 3

We would express our thanks to the reviewer for reviewing this manuscript and helpful comments.

A) The reviewer requested the addition of the paragraph about the effects of aqueous-extracts of ginger. Therefore, we added their paragraph and literatures such as Alsherbiny *et al.* (2019), and clarified the purpose of this study in Introduction **(Page 2, lines 57–64)**, and Discussion **(Page 6, line 269)**.

B) The reviewer requested the discussion that $PGE_2$ production was reduced in macrophages and inflammatory sites but increased in gastric ulcer region. Therefore, we added their paragraph **(Page 8, line 366 – Page 9, line 373)**.

C) As the comments of the reviewer, We added 3D-HPLC profiles of shokyo and kankyo **(Supplemental Figure 1)** and related sentence **(Page 2, lines 69–70, Page 7, lines 298–299)**, and changed thereafter figure numbers.

D) As the comments of the reviewer, we added the paragraph about the limitation of this study and the ideas for future research **(Page 9, lines 374–378)**.

---

## Round 0.3 · accepted · Accept

Thank you for addressing the last requests!

---

## Author Rebuttal · Round 0.3

August 19, 2019
Dear Pedro Silva, Academic Editor:

Thank you for your e-mail of August 12, with regard to our manuscript (Article ID: 36484) entitled "Effects of shokyo (*Zingiberis Rhizoma*) and kankyo (*Zingiberis Processum Rhizoma*) on prostaglandin $E_2$ production in lipopolysaccharide-treated mouse macrophage RAW264.7 cells" by Toshiaki Ara *et al*. Moreover, we would express our thanks to you again and reviewer 2 for reviewing this manuscript and helpful comments. I am sending therewith our revised manuscript with changes indicated by using red color and figures, with sheet detailing our response to the points raised and the changes we have made.

I hope you will find the data interesting and would consider whether this manuscript is acceptable for publication in PeerJ.

Thank you for your kind consideration.

Sincerely yours,

Toshiaki Ara, D.D.S., Ph.D.

Department of Pharmacology

Matsumoto Dental University

1780 Gobara Hirooka, Shiojiri, Nagano 399-0781, Japan

Phone: 81-263-51-2103

FAX: 81-263-51-2102

E-mail: toshiaki.ara@mdu.ac.jp

Answers to comments of Reviewer 2

We would express our thanks to the reviewer for reviewing this manuscript and helpful comments.

1) As the reviewer pointed out, we have misunderstood the reviewer's comment. Because the time schedule of "sequential treatment" in the legends in Figure 3A were right but the schema was wrong, we corrected the schema (LPS treatment for 6 h and thereafter with each herb for 1 h).

2) Image analysis software ImageJ (made by NIH, and originally named NIH Image) is widely used in many studies to analyze the intensity of bands on X-ray films (although the method using ImageJ is omitted in many studies). We describe the several examples of the sentence using ImageJ in Materials and Methods section as below.

- "the blots were analysed using Image J software."
  - Molecules 23: 3319 (2018)
- "Densitometric analysis was performed using NIH Image J Software."
  - Journal of Pharmacological Sciences 133: 18–24 (2017)
- "the bands were analyzed using ImageJ software."
  - Oncotarget 8: 42001–42006 (2017)
- "Protein levels were quantified using Image J software."
  - Mediators of Inflammation 2015: 329405 (2015)
- "the optical density of an equal area for each band was determined using Image J software."
  - Biological Pharmaceutical Bulletin 34: 1864–1873 (2011)

We also modified the sentence concerning ImageJ (**Page 4, lines 141–143**).

3) According to reviewer's comment, we replaced with the new Western blotting results without splicing (Figure 6C).